# The Evolution of Rag Gene Enhancers and Transcription Factor E and Id Proteins in the Adaptive Immune System

**DOI:** 10.3390/ijms22115888

**Published:** 2021-05-31

**Authors:** Genki Yoshikawa, Kazuko Miyazaki, Hiroyuki Ogata, Masaki Miyazaki

**Affiliations:** 1Bioinformatics Center, Institute for Chemical Research, Kyoto University, Uji 611-0011, Japan; yoshikawagenki@gmail.com; 2Laboratory of Immunology, Institute for Frontier Life and Medical Sciences, Kyoto University, Kyoto 606-8507, Japan; kamiyazaki@infront.kyoto-u.ac.jp

**Keywords:** adaptive immune system, Rag1/2 gene enhancers, E protein, Id protein, bHLH transcription factors, T and B cell development

## Abstract

Adaptive immunity relies on the V(D)J DNA recombination of immunoglobulin (*Ig*) and T cell receptor (*TCR*) genes, which enables the recognition of highly diverse antigens and the elicitation of antigen-specific immune responses. This process is mediated by recombination-activating gene (Rag) 1 and Rag2 (Rag1/2), whose expression is strictly controlled in a cell type-specific manner; the expression of *Rag1/2* genes represents a hallmark of lymphoid lineage commitment. Although *Rag* genes are known to be evolutionally conserved among jawed vertebrates, how *Rag* genes are regulated by lineage-specific transcription factors (TFs) and how their regulatory system evolved among vertebrates have not been fully elucidated. Here, we reviewed the current body of knowledge concerning the cis-regulatory elements (CREs) of *Rag* genes and the evolution of the basic helix-loop-helix TF E protein regulating *Rag* gene CREs, as well as the evolution of the antagonist of this protein, the Id protein. This may help to understand how the adaptive immune system develops along with the evolution of responsible TFs and enhancers.

## 1. Introduction

Our body is protected from invading pathogens by immune responses, which are primarily mediated by two distinct types of cells, the adaptive and innate immune cells. These cells cooperatively function to induce inflammatory responses to eliminate pathogens from the body. Adaptive immune cells, such as T and B cells, elicit pathogen-specific immune responses through the recognition of specific antigens, while innate immune cells, including macrophages, neutrophils, dendritic cells, and histiocytes, are activated by pattern recognition receptors (PRRs), which recognize distinct microbial components. The adaptive immune system (AIS) relies on the assembly of T cell receptor (*TCR*) and immunoglobulin (*Ig*) genes from arrays of variable (V), diversity (D), and joining (J) gene segments. These assembled antigen receptors are able to recognize highly diverse antigens and elicit antigen-specific immune responses [1,2]. This V(D)J recombination of *TCR* and *Ig* genes is mediated by recombination-activating gene 1 (Rag1) and Rag2 protein complex, and Rag1/Rag2 (Rag1/2) complex recognizes and cleaves recombination signal sequences (RSSs) flanking the TCR and Ig V, D, and J gene segments [3]. Rag1 primarily binds and cleaves DNA, while Rag2 enhances Rag1 binding and is a vital co-factor for DNA cleavage [3]. Because the *Rag1/2* genes are exclusively expressed in T cell and B cell progenitor/precursor stages, their expression implies adaptive lymphoid lineage commitment [4]. *Rag* genes are known to be evolutionally conserved among jawed vertebrates. The current model suggests that *Rag1/2* genes evolved from the ancestral RAG transposase genes [5,6]. The discovery of ProtoRag in the cephalochordate amphioxus strongly support this model [7], and a recent report presenting cryo-electron microscopy structures of RAG and ProtoRAG transposase shows the mechanism underlying the properties of RAGs with appropriate DNA cleavage and transposition activities [8]. They identified two adaptations specific to jawed vertebrates in RAG1 and RAG2 that suppress RAG-mediated transposition [8].

After T cell lineage commitment in the thymus, TCRβ V(D)J rearrangement is initiated at immature CD4^−^CD8^−^ (double negative; DN) cells (T progenitor cell; pro-T cell), and DN cells are further divided into DN1-4 populations. Following the selection of TCRβ in DN3a cells, DN3a cells start proliferation and differentiate into DN3b-DN4 stage and further CD4^+^CD8^+^ (double positive; DP) cells (T precursor cell; pre-T), in which TCRα VJ rearrangement is performed [9]. As well as T cell, upon B cell lineage commitment from common lymphoid progenitors (CLPs) in the bone marrow, V(D)J recombination of Ig heavy chain (*Igh*) and light chain (*Ig**κ* and *λ*) occurs in B cell progenitor (pro-B) and precursor (pre-B) cells, respectively [10,11]. CLPs give rise to adaptive lymphoid cells (T and B cell), innate lymphoid cells (ILCs), and plasmacytoid dendritic cells (pDCs). T and B cell lineage commitments are instructed by a set of lineage-specific transcription factors’ (TFs’) expression: E2A, Ebf1, Foxo1, and Pax5 for B cell and E2A/HEB, Gata3, Tcf1, Bcl11b, Runx, Ikaros, and Pu.1 for T cell [12,13,14,15,16]. Notably, ILCs and T cells show functional similarities in cytokine production, and they commonly express Bcl11b, Tcf1, Gata3, and Runx during their development and activation [17]. What TFs drive adaptive lymphoid lineages? Importantly, E2A and HEB act in synergy to establish T cell identity. In addition, they simultaneously suppress the aberrant ILC development in thymus [18]. Similarly, Ebf1 and Pax5 determine the B cell lineage by repressing genes leading to the T cell and ILC lineages [19]. These adaptive lymphoid lineage-specific TFs were also expected to regulate *Rag1/2* gene expression to drive the difference between adaptive and innate immune cells. Lineage-specific TFs alter gene expression patterns by binding to specific DNA sequences within cis-regulatory elements (CREs), and TF bindings modulate the enhancer–promoter interactions. Many previous studies have attempted to identify the CREs associated with *Rag1/2* gene expression [4,14]. Although both developing T and B cells express the *Rag1/2* genes, the *Rag* gene enhancers differ in T and B cells. In T cells, an anti-silencer element (ASE), which is located 73 kb upstream of the *Rag2* gene and is 8 kb in length, is essential for *Rag1/2* gene expression in developing T cells, but not in developing B cells [20]. Recently, a T cell-specific *Rag* enhancer region (*R-TEn;* 1 kb) was identified within the ASE region, which is regulated by E2A [21]. In contrast, two B cell-specific enhancers (*R1B* and *R2B* (overlapping with *Erag*)) are critically required for *Rag1/2* expression in pro-B cells, and the deletion of both regions results in developmental arrest at the pro-B stage (Figure 1) [21,22]. These results raised questions regarding what lineage-specific TFs regulate these CREs to determine the adaptive lymphoid lineage and how these TFs and CREs have evolutionarily developed during the evolution of AIS. In this report, we reviewed the role of adaptive lymphocyte-specific TFs, especially the transcriptional balance between the E and Id proteins (E-Id axis) [23], and the mechanism by which E2A regulates *Rag1/2* expression, and the evolution of E-Id proteins and *Rag1/2* enhancers among species.

## 2. Regulation of *Rag1/2* Gene by T or B Cell-Specific Enhancers

### 2.1. CREs for the Rag Gene, and Lineage-Specific Transcription Factors

*Rag1/2* gene expression is stringently controlled in a cell type-specific manner. The first wave of RAG expression is required for the recombination of *TCR**β* and *Igh* genes in pro-T and pro-B cells, respectively. After the β-selection of pro-T cells and pre-BCR selection of pro-B cells, *Rag* expression is transiently downregulated during the developmental transition toward precursor stages (DP and pre-B cells). In precursor cells, *Rag1/2* genes are upregulated for the VJ recombination of *TCR**α*, *Ig**κ,* and *λ* genes. Following successful *TCR* and *Ig* gene recombination, *Rag1/2* gene expression is completely suppressed in mature T and B cells [10,14,24,25]. Both in mouse and human, impairment or loss of *Rag* gene expression and functions results in severe combined immunodeficiency, resulting from developmental arrest at pro-T and pro-B cell stages [26,27]. Furthermore, persistent *Rag* expression led to profound immunodeficiency in mouse [28]. Thus, *Rag1/2* genes are stringently regulated during T and B lymphopoiesis. There have been in vivo and in vitro studies that attempted to identify the CREs associated with *Rag1/2* expression [4]. Deletion of *Erag*, which is 23 kb upstream of the *Rag2* gene, caused impaired *Rag1/2* expression in pro-B cells and a moderate developmental block at the pro-B stage but did not affect the *Rag* gene expression in T cell development [22]. Interestingly, deletion of two B cell-specific enhancers, *R1B* and *R2B* (partially overlapping with *Erag*), resulted in a developmental arrest at the pro-B stage, indicating the enhancer redundancy of *Rag* gene in B cells [21]. On the other hand, the T cell-specific *Rag* gene enhancer *R-TEn*, which is included in ASE, is critically important for *Rag1/2* expression and TCR recombination during thymocyte development, and its deletion led to a developmental block at the DN3 cells and DP cells in fetal and adult thymus without affecting B cell development [20,21,29]. Taken together, *Rag* gene expression is tightly regulated in a cell type-specific manner, and T and B cells use distinct enhancer regions for *Rag* gene expression.

Previous reports and ChIP-seq showed that many T cell- or B cell-specific TFs bind to these enhancer regions (T cells: Tcf1, Bcl11b, Gata3, Runx1, Satb1, and Ikaros; B cells: Pax5, Ebf1, Foxo1, Ets1, Irf4, and Ikaros) [21,29,30,31]. What TFs regulate these cell type-specific enhancers for *Rag* genes? Notably, mutating the E-box motifs in the *R-TEn* enhancer (*R-TEn-E-box-mutant*) in mice, in which the binding of E2A is blocked, resulted in an impaired *Rag1/2* expression and blocked thymocyte development, as well as deletion of the entire *R-TEn* region. Of note, the *R-TEn-E-box-mutant* abolished chromatin accessibility throughout the entire *Rag* gene locus. These results indicated that E-protein binding to the T cell-specific *Rag* gene enhancer is required for T cell-specific spatial interactions to enhance *Rag1/2* expression [14,21]. Notably, blocking E2A binding to the *Rag1* gene promoter region (*R1pro*) by generating E-box motif mutations alone resulted in the complete loss of *Rag1* expression without affecting *Rag2* expression in both developing T and B cells, leading to developmental arrest at the pro-T and pro-B cell stages [21]. Taken together, these results strongly suggest that the activities of T cell-specific enhancer and *Rag1* promoter depend on the binding of E2A to these regions and that E2A is a core TF that specifies the adaptive lymphoid cell identity through the regulation of *Rag* gene expression.

### 2.2. Evolution of Rag Gene Enhancer

Enhancer regions play a crucial role in precise pattern and amounts of gene expression during development, and divergence of the DNA sequence within enhancer region is considered to be related to the phenotypic variations among species [32]. This suggests that the phylogenetic conservation of DNA sequences within *Rag* gene enhancers reflect the evolution of *Rag* gene regulation. Although *Rag1/2* genes are well known to be conserved among jawed vertebrates, the conservation of *Rag* gene enhancers had not been investigated. Thus, we investigated the conservation of *R-TEn*, *R1B,* and *R2B* regions and E-box motifs in these regions [21]. We found that DNA sequence similarities in *R-TEn* and *R2B* are readily observed among mammals, most birds, and reptiles; however, sequence similarities of these enhancers are not noticeable in the corresponding genomic regions of amphibians and fishes (Figure 2) [21]. Furthermore, we observed significant conservation of E-box motifs in conserved *R-TEn*, *R2B,* and *R1B* regions (Figure 2) [21]. These results show the discordance in the conservation of *Rag* genes and their enhancers among jawed vertebrates and the possibility of divergent cis-regulatory modules of *Rag* genes in terrestrial animals, aquatic animals, and amphibians. Thus, we proposed that terrestrial animals evolutionarily acquired the E protein-mediated regulatory mechanisms as enhancers to increase the *Rag* gene expression, which induce higher expression of *Rag genes* and enable a diverse range of *TCR* and *Ig* gene recombination to protect our bodies from a wide range of pathogens.

Regarding the evolution of AIS among vertebrates, cytidine deaminases CDA1 and CDA2 in jawless vertebrates are counterparts of *Rag1* and *Rag2* in jawed vertebrates and evolutionarily developed AIS as genome editors [33,34,35]. Furthermore, the recombination of *Ig* and *TCR* in fish seems to be more diverse than that in mammals, for example, the plasticity of T/B cells and the repertoire usage of *TCR* and *Ig* [36]. Given that the locations of B cell development among birds, reptiles, amphibians, and fish are different, it is reasonable that the variation in enhancer regions among species produces diversification of *Rag1/2* gene regulation, such as timing. Considering this, it is surprising that both enhancer and promoter activities are critically controlled by E protein binding.

## 3. E Proteins and Id Proteins in Adaptive Lymphocyte Development

E proteins are basic helix-loop-helix (bHLH) transcription factors involved in multiple developmental processes. E proteins include E12, E47, E2-2 (TCF4), HEB (TCF12), daughterless (Da), and HLH-2 [23]. E proteins bind as homodimers or heterodimers to the E-box motif (CANNTG) within enhancer regions of their target genes. The mammalian E protein family plays important roles in hematopoiesis. However, the *Drosophila* gene product (da) and *Caenorhabditis elegans* gene product HLH-2 are involved in other developmental pathways. Da is essential for both neurogenesis and sex determination in *D. melanogaster* embryonic development [37]. HLH-2 is required for the development and function of the regulatory cells of the *C. elegans* somatic gonad [38]. Id proteins contain an HLH domain missing the basic region that is essential for specific DNA binding and form heterodimers with bHLH proteins such as E proteins [39]. When the Id protein forms heterodimers with the E protein, the Id protein antagonizes the DNA binding of E proteins and functions as a negative regulator [40]. Id proteins include Id1-4 and the *D. melanogaster* gene product *extramacrochaete* (*emc*) [23].

It is well established that a majority of adaptive lymphocyte development trajectories require regulation by E and Id proteins [23,39,41]. *E2A* (*Tcf3*) is critically required for B cell lineage commitment [42,43] and the E2A gene encodes E12 and E47 proteins, which are generated by differential splicing [44,45]. In lymphoid progenitor cells, E2A orchestrates the B cell fate, along with Ebf1, Foxo1, and other TFs [12,46]. Upon T cell lineage commitment, E2A and HEB act in synergy to establish T cell identity and to suppress ILC development [18]. Likewise, HEB plays a role in iNKT cell development [47], and E2A and HEB also play important roles in the positive selection of DP thymocytes [48]. Interferon-producing plasmacytoid dendritic cell (pDC) development is controlled by E2-2, whereas antigen-presenting classical dendritic cells (cDCs) are orchestrated by Id2 through neutralizing E2-2 activity [49,50,51].

*Id2* and *Id3* are involved in both T and B cell development to modulate E protein DNA binding activity. *Id2* is particularly important for ILC, NK, and LTi cell development through the suppression of E protein activity [13,52]. In B cell development, Id3 is induced in response to TGFβ signaling for survival during early B cell development [53]. Id3 is highly expressed in naïve mature B cells and downregulated in activated germinal center B (GCB) cells, while E2A protein abundance is low in naïve B cells but high in GCB cells to induce AID expression in cooperation with E2-2 [54,55]. In T cell development, *Id3* is first upregulated by pre-TCR signaling in DN3 cells and further upregulated upon positive selection of TCR signaling in DP cells [56,57]. Similarly, γδ TCR-mediated signaling induces high levels of *Id3* abundance and E2A-Id3-Tcf1 transcription axis control γδ T cell development and effector function [58,59]. Furthermore, *Id3* plays a key role in follicular helper T (T_FH_) and follicular cytotoxic T(T_FC_) cell development through the regulation of CXCR5 expression [60,61,62]. Notably, *Id2* and *Id3* have distinct roles in the differentiation of CD8 cytotoxic T cells toward effector and memory T cells [63,64]. In addition, regulatory T (Treg) cells also critically require *Id2* and *Id3* expression to suppress systemic T_H2_ inflammation and function as a gatekeeper for follicular regulatory T (T_FR_) cell [65]. Taken together, the E–Id protein axis controls the adaptive lymphocyte development and activation to maintain immunological homeostasis.

## 4. Evolution of E and Id Proteins

In this section, we address the question of how the E–Id axis was evolutionally developed. Emc is a negative feedback regulator that prevents runaway self-stimulation of *Da* gene expression in *Drosophila*. Coupled transcriptional feedback loops maintain the widespread Emc expression that restrains Da activity to induce neurons [66], suggesting that the transcriptional regulation system by E and Id proteins is conserved from the common ancestor of mammals and *Drosophila*.

Three E protein homologs and two Id protein homologs were found in the lamprey (*Petromyzon marinus*) (Figure 3). A reconstructed maximum likelihood phylogenetic tree of E protein homologs indicates that homologs of jawed vertebrates form three clades for E2A, E2-2, and HEB. Lamprey E protein homologs are located outside these three clades of E proteins of jawed vertebrates, although their positions are not well resolved (Figure 3A, Appendix A). A reconstructed maximum likelihood phylogenetic tree for Id protein homologs indicates that homologs of jawed vertebrates form four clades corresponding to Id1 to Id4 (Figure 3B, Appendix A). Lamprey Id protein homologs form a clade with Id2 of jawed vertebrates, although this position is not statistically supported. These multiple clades of E and Id proteins conserved in jawed vertebrates strongly suggest that these paralogs were generated through the widely recognized two rounds of whole genome duplication (WGD) in vertebrates [67]. It is plausible that ancestral jawed vertebrates probably had four paralogs for each of the E and Id proteins, and one of the four E protein paralogs was lost early in evolution prior to the divergence of jawed vertebrates. Recent research proposed that all extant vertebrates share the first duplication, which occurred in the Cambrian, and the second duplication is found only in jawed vertebrates and occurred in the Ordovician [67]. Given the unstable phylogenetic positions of lamprey E and Id homologs, it is unclear from the current data whether this evolutionary scenario could also explain the existence of paralogs in jawless vertebrates.

## 5. Discussion

Jawed vertebrates (gnathostomes) possess AIS that can recognize and initiate a protective response against invading pathogens. AIS in jawed vertebrates is centered on T and B lymphocytes bearing TCR and B cell receptors (BCRs, Igs), which are generated through the V(D)J recombination mediated by Rag1 and Rag2 [3]. The TCRs recognize peptide fragments of antigens complexed with molecules encoded by the major histocompatibility complex (MHC) class I and class II genes. Therefore, *Ig*, *TCR*, *Rag1* and *Rag2,* and *MHC* class I/II genes have an integral role in AIS in jawed vertebrates. Homologs of these genes have been identified in all extant classes of jawed vertebrates [71]. On the other hand, jawless vertebrates (agnathans) possess lymphocyte-like cells (LLCs) that morphologically resemble the T and B cells of jawed vertebrates [72,73,74]. Many of the genes encoding transcription factors involved in AIS in jawed vertebrates are also encoded in the genome of jawless vertebrates [75]. However, sequence and transcriptome analysis of jawless vertebrates provided no evidence for the presence of *Ig*, *TCR*, *Rag1/2*, or *MHC* genes [72,76,77]. Instead of *TCR* and *Ig*, different types of antigen receptors, which are known as variable lymphocyte receptors (VLRs), composed of highly diverse leucine-rich repeat (LRR) modules, have been identified in lampreys, and, like Rag-mediated recombination of *TCR* and *Ig* in jawed vertebrates, VLRs are generated by cytidine deaminase (CDA) 1 and 2 [33,34,35,73,78]. Interestingly, CDA2 is closely related to activation-induced cytidine deaminase (AID), which is essential for class-switch recombination (CSR) and somatic hypermutations (SHMs) in human and mice [34,79]. The similarities of multiple components of AIS between jawed and jawless vertebrates suggest that the roots of the system originated in the common ancestor of all vertebrates [80]. Additionally, a primitive AIS emerged already in the common ancestor of all vertebrates, although the possibility of independent acquisitions of the AIS in jawed and jawless vertebrates cannot be excluded. The *Rag1* and *Rag2* genes that are essential for AIS in jawed vertebrates are proposed to have arisen from a transposable element and transmitted vertically through chordate and jawed vertebrate evolution [6,7,81]. This scenario suggested that the acquisition of novel molecular capabilities was a crucial event in the evolution of AIS in jawed vertebrates. Of note, we have recently demonstrated that E2A regulates enhancer/promoter activity of *Rag* gene during T and B cell development [21] and Id2 is essential for innate lymphocytes, including dendritic cells, suggesting that the E–Id axis determines the cell fate of lymphocytes between adaptive and innate immune cells [18,23]. Therefore, it seems that the existence of paralogous E and Id genes at the time of the horizontal acquisition of transposable *Rag* genes contributed to the genesis of the AIS in jawed vertebrates. The two rounds of WGD are considered important for the acquired immunity of jawed vertebrates [82,83]. Previous research proposed that two rounds of WGD played a major role in the duplication of many signaling genes ancestrally used in nervous system development and function that were later co-opted for new functions during evolution of the AIS [84]. E-Id paralogous genes represent new evidence that paralogous genes arisen by WGDs could be co-opted for new functions in AIS in jawed vertebrates. In fact, functional redundancies among E proteins or Id proteins are observed in T cell lineage commitment and development (E2A and HEB, and Id2 and Id3), Treg cell function (Id2 and Id3), and germinal center B cell development (E2A and E2-2), suggesting that the duplication of E or Id genes could give rise to diversity and stability in AIS [48,60,65,85,86]. These observations also reinforce the hypothesis that not only the acquisition of novel molecular capabilities but also the co-option and redirection of preexisting systems are the major sources of innovation [87].

## Figures and Tables

**Figure 1 ijms-22-05888-f001:**
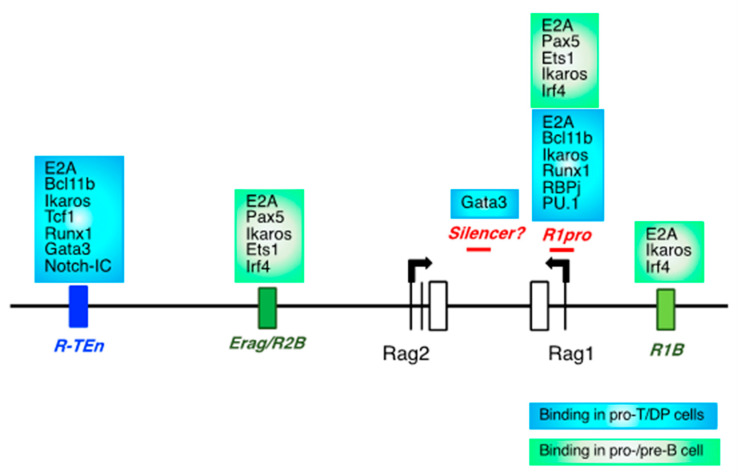
*Rag* gene enhancers in developing T and B cells. A schematic diagram of *Rag* gene locus and enhancers in developing T and B cells. The blue box indicates T cell-specific enhancer (R*-TEn*) and TF binding in pro-T and DP cells. The green boxes indicate B cell-specific enhancers (*R1B* and *R2B*) and TF binding in pro-/pre-B cells. The red lines indicate the *Rag1*-promoter region (*R1pro*) and silencer (*Silencer*) of *Rag* gene, respectively.

**Figure 2 ijms-22-05888-f002:**
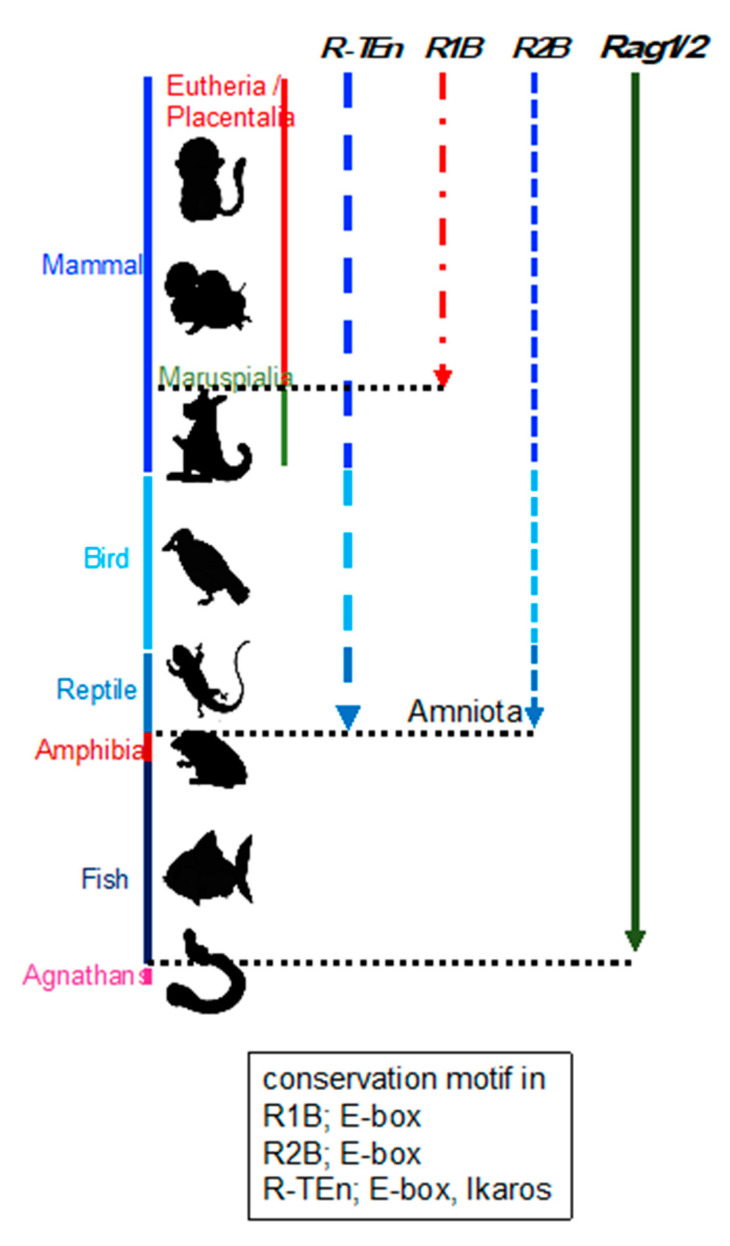
Schematic summary of the conservation of *R-TEn*, *R1B*, and *R2B* among vertebrates. Black, dotted lines indicate the border between placentaria and maruspialia, reptile and amphibia, and fish and agnathans. The conserved motifs in each enhancer region are shown in the box [21].

**Figure 3 ijms-22-05888-f003:**
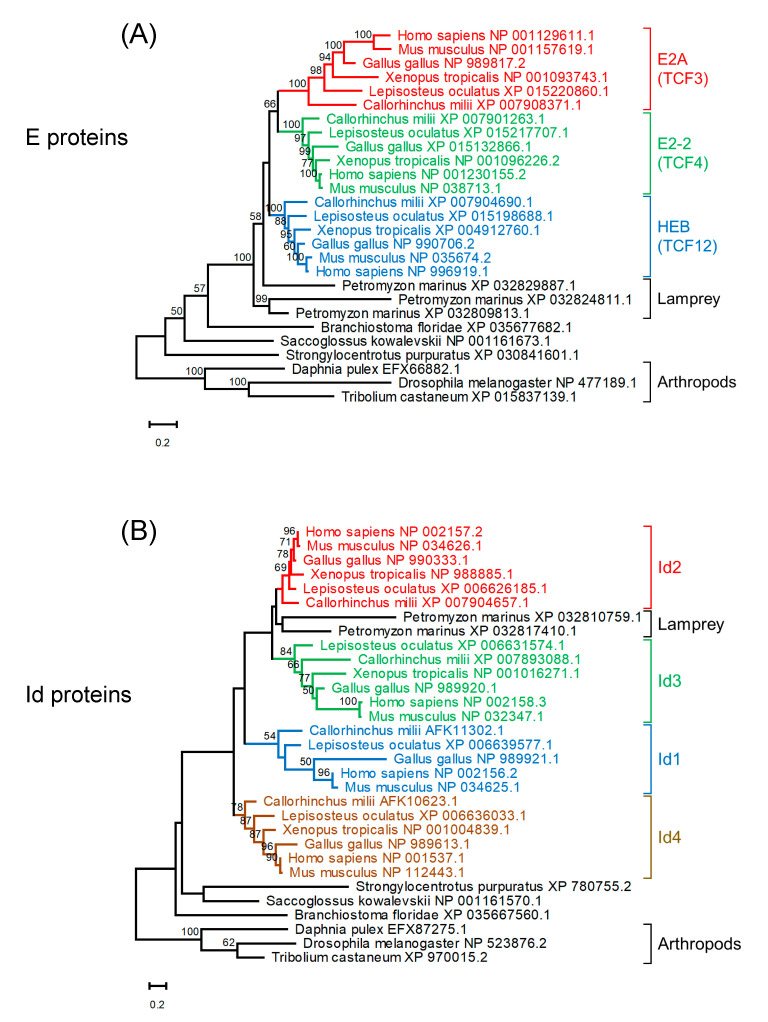
Maximum likelihood phylogenetic trees of homologs of E proteins (**A**) and Id proteins (**B**). Sequences were aligned using MAFFT (v7.453) [68] with default parameters. Tree reconstruction was performed using RAxML (version 8.2.12) [69] with the JTT + F substitution model and PROTGAMMA parameter with 100 bootstrap replicates. Phylogenetic trees were visualized using MEGA-X (version 10.2.4) [70]. Bootstrap values are given along the branches.

## Data Availability

Not applicable.

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
