# Peer review of "The Evolution of Rag Gene Enhancers and Transcription Factor E and Id Proteins in the Adaptive Immune System"

_ijms, 2021, doi:10.3390/ijms22115888_

Round 1

Reviewer 1 Report

The review by Yashikawa et al. provides a nice summary of the current understanding of T and B lineage-specific Rag gene enhancers and transcriptional factors regulating these enhancers, as well as the evolution of these enhancers and transcriptional factors. The review is well written and was a pleasure to read.

I have a few points that should be addressed prior to publication.

  1. The authors discussed how E proteins bind to Rag gene enhancer regions and regulate Rag gene expression. The authors then discussed how E-Id protein axis controls lymphocyte development and function. However, it is not clear whether Id proteins modulate E protein activities to affect Rag gene expression during T/B cell development.
  2. An additional figure would be appreciated to depict the CREs of Rag gene and the binding of transcription factors.
  3. Line 51, “right-chain” à “light-chain”.
  4. Line 53-54, I wonder if it would be more accurate to state “plasmacytoid dendritic cells” instead of the broad term “dendritic cells”, as it seems it is rather controversial whether CLP can give rise to cDC or not (PMID: 32908299).

Author Response

Overall, we really appreciate the reviewer’s comments. These are productive and thoughtful comments.

We provide point-by-point responses to reviewer’s comments.

  1. The authors discussed how E proteins bind to Rag gene enhancer regions and regulate Rag gene expression. The authors then discussed how E-Id protein axis controls lymphocyte development and function. However, it is not clear whether Id proteins modulate E protein activities to affect Rag gene expression during T/B cell development.

Yes, I totally agreed that there was not clear data whether Id-proteins modulate to suppress Rag gene expression by antagonizing E-protein activities. As we reported, Id3 is upregulated by pre-TCR signaling and TCR signaling in b-selection and positive selection, and the deletion of both Id2 and Id3 resulted in the defect of positive selection of the conventional T cell (Genes & Development 2015). Although E2A/E-protein binding to the Rag gene enhancer is critical for the Rag gene expression, however, we do not have any data that Id-deficiency results in the prolonged Rag1/2 expression after positive selection.

Therefore, it is not clear whether Id-protein suppress the Rag gene expression in the physiological condition, not in the overexpression of Id-protein.

  1. An additional figure would be appreciated to depict the CREs of Rag gene and the binding of transcription factors.

Thank you for your suggestion. We added a schematic diagram of Rag gene locus including Rag gene CREs, instead of VDJ recombination.

  1. Line 51, “right-chain” à “light-chain”.

Thank you so much. It was a big mistake.

  1. Line 53-54, I wonder if it would be more accurate to state “plasmacytoid dendritic cells” instead of the broad term “dendritic cells”, as it seems it is rather controversial whether CLP can give rise to cDC or not (PMID: 32908299).

We appreciate your comment. I agree with you. I changed the sentence (line 59).

Reviewer 2 Report

In this manuscript, the authors review the current (limited – therefore a review is very welcome) knowledge on the regulation of RAG1 and RAG2 expression in the context of evolution in a wide range of species beyond mammals only.

As the authors mention, RAG is a critical protein that performs V(D)J recombination at the chromosomal level to produce recombined BCR or TCR genes, and is to be tightly regulated to be expressed only in B and T cell progenitors at the good stage of development (RAG targets being very deleterious).

The authors have recently published the discovery of new enhancers elements for RAG expression (Science immunology 2020) and wrote a review about chromatin regulation of RAG expression (Frontiers Immunology 2021). Therefore, the authors are most suited and trustable to write this manuscript, and the evolutionary insight developed here is very complementary to the two above-mentioned papers.

Overall, the review is well written and details the enhancer regions controlling RAG expression and which transcription factors have been linked with both B- or T- cells development and control of RAG expression in a comprehensive manner for readers that did not know previously of such mechanisms. Further, the authors put in context RAG expression in unconventional populations (gamma-delta T cells) and with Somatic Hyper Mutation, which is well appreciated.  I am therefore very positive about publishing this manuscript.

However, there is only one point that made me suggest major revisions: I think the figures are not informative enough at this stage and do not help enough to understand the text, and their caption needs to be explained more. I give examples / suggestions how to solve this point. The text is already well written and informative enough.

Major points

  • Can the authors introduce to the reader the difference between RAG1 and RAG2 function? The gene locus organization of RAG1 and RAG2 should be visually explained to the reader in a figure (in particular to illustrate line 68-69 where the authors say T- and B- cell RAG promoter usage is different).
  • Figure 1 is a very classical illustration of VDJ recombination that can be found in many papers and is not really informative here. Can the authors add information related to RAG regulation, or maybe remove this figure?
  • Figure 2, since the focus is the evolution of the enhancers of RAG, it would be important to show the sequence alignment of such promoter regions.
  • Figure 3, Could the authors explain in the caption how the tree was made (for instance how the bootstrap was performed?) and also show some alignments, for instance in relation with known DNA binding domains if they are known?

Minor points:

  • Number for affiliations should be as exponent
  • Lines 44-45, I didn’t understand the meaning of this sentence (what is the mechanism suppressing transposition?) - maybe explain
  • Line 48: the authors could precise B selection is happening between DN3a and DN3b stages, if they like
  • Line 184, typo, the two reference numbers can be merged
  • By reading the manuscript, I got a few questions, which the authors could discuss if they like, but which could be also be out of context and ignored. Are there known factors that mediate RAG downregulation / suppression after VDJ recombination? And is there evidence for requiring not only one E-protein but a transcriptional complex of one E-protein and another transcription factor (that could be activator or repressor) to mediate RAG expression for instance?

Author Response

Thank you so much for reviewer’s comments. These are very helpful and thoughtful.

We provide point-by-point responses to reviewer’s comments.

  1. · Can the authors introduce to the reader the difference between RAG1 and RAG2 function? The gene locus organization of RAG1 and RAG2 should be visually explained to the reader in a figure (in particular to illustrate line 68-69 where the authors say T- and B- cell RAG promoter usage is different).
  • Figure 1 is a very classical illustration of VDJ recombination that can be found in many papers and is not really informative here. Can the authors add information related to RAG regulation, or maybe remove this figure?

Thank you for this suggestion. We remove VDJ recombination, instead of this, we added a schematic diagram of Rag gene locus including Rag gene enhancers. We believe that this may help the readers understand it.

  1. Figure 2, since the focus is the evolution of the enhancers of RAG, it would be important to show the sequence alignment.

Indeed, we analyzed and showed the sequence alignment among species in recent paper (Miyazaki, et al., Science Immunology, 2020; Figure 7 and figure S7A-C). In the previous version of the manuscript, we did not mention that the DNA sequence similarities are shown in this paper. Thus, we changed the sentence in revised version (line 147-149).

Figure 3, Could the authors explain in the caption how the tree was made (for instance how the bootstrap was performed?) and also show some alignments, for instance in relation with known DNA binding domains if they are known?

Thank you for these suggestions. We added the sentences in figure legends for explanation. In addition, we showed the multiple sequence alignments of E-proteins and Id-proteins in supplementary figure.

Lines 44-45, I didn’t understand the meaning of this sentence (what is the mechanism suppressing transposition?) - maybe explain

We added more sentences in line 46-49.

Line 48: the authors could precise B selection is happening between DN3a and DN3b stages, if they like

We added some sentences to explain this (line 51-54).

Line 184, typo, the two reference numbers can be merged

Thank you so much for your pointing out, I corrected this and other sites of similar mistakea.

Are there known factors that mediate RAG downregulation / suppression after VDJ recombination?

This is a very important question. We believe that blocking E-protein binding to the Rag gene enhancer region immediately lose the accessibility in this region and results in the disruption of super-enhancer formation. Therefore, Id3 induction mediated by pre-TCR and TCR signaling is able to down-regulation the Rag gene expression.

And is there evidence for requiring not only one E-protein but a transcriptional complex of one E-protein and another transcription factor (that could be activator or repressor) to mediate RAG expression for instance?

This is what I really want to know. E-protein binding to the enhancer region is critically required for the maintenance of the chromatin accessibility in enhancer regions, but it does not mean that E-protein binding itself could induce target gene expressions. Therefore, we have attempt to identify the collaborating TFs to induce the target genes associated with T and B cell specification. Now, I found the interesting TF which acts together with E2A in both T and B cell commitment. We are looking forward to unraveling this molecular mechanism.

There are some previous reports of repressor TF which interacts with E2A, but I do not trust this. We will clarify the molecular mechanisms of E-protein mediated gene regulation. I really appreciate these questions.

Round 2

Reviewer 2 Report

The authors have answered all the previous points (in particular about the figures). They have reshaped figure 1, added a new supplementary figure with alignments and explained the methods for generating the trees. This is very much appreciated!

I would like to pinpoint the following paper to the authors that got published a few days ago: https://www.biorxiv.org/content/10.1101/2021.05.13.443498v1 "Lung Epithelial Cells Can Produce Antibodies Participating In Adaptive Humoral Immune Responses"  - I actually don't believe that RAG could be expressed on epithelial cells. If the authors wish to discuss this in the manuscript, they can, but this is not necessary - the manuscript is publishable as it is.